# Anion-Complexation-Induced Emission Based on Aggregation-Induced Emission Fluorophore

**Dongxing Ren, Liangliang Zhang, Hongwei Qian and Tangxin Xiao ***

Jiangsu Key Laboratory of Advanced Catalytic Materials and Technology, School of Petrochemical Engineering, Changzhou University, Changzhou 213164, China
* Correspondence: xiaotangxin@cczu.edu.cn

**Abstract:** Aggregation-induced emission (AIE) materials have attracted increasing research interest in recent years due to their excellent fluorescence properties in an aggregated state. Concurrently, anion coordination interactions have played a key role in the development of supramolecular assemblies and sensors. In the past decade, investigations towards fluorescent materials or sensors based on AIE and anion coordination interactions are continuously being reported. In this minireview, we briefly summarize the burgeoning progress of AIE-based materials and sensors driven by anion coordination interactions. We believe that an increasing number of achievements in anion-coordination induced emission materials will appear in the near future and will demonstrate potential applications, including bio-imaging and bio-sensors.

**Keywords:** supramolecular chemistry; aggregation-induced emission; anion coordination; fluorescent sensor; self-assembly

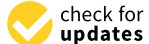



## 1. Introduction

Luminescent materials based on organic fluorophores have attracted tremendous attention in recent decades owing to their extensive applications in bio-imaging, sensing, and optoelectronics [1–4]. Conventional organic fluorophores, such as pyrene, perylene, coumarin, and fluorescein, are planar-shaped and highly emissive in a dilute solution. These fluorophores usually undergo an aggregation-caused quenching (ACQ) behavior in the aggregated state, greatly limiting their potential in real applications. Taking perylene as an example, as shown in Scheme 1a, when perylene is dissolved in a good solvent such as tetrahydrofuran (THF) at a low concentration (such as $2 \times 10^{-5}$ M), the solution exhibits a strong fluorescence. However, upon the addition of a poor solvent such as $H_2O$, the perylene molecules gradually undergo a disordered aggregation. When the fraction of $H_2O$ is increased to 80%, the fluorescence of the perylene is quenched due to strong aggregation. At the beginning of this century, Tang and colleagues developed another type of organic fluorophores which exhibit no emission in a dilute solution but exhibit strong emission in an aggregated state, contrary to the ACQ fluorophores [5,6]. This kind of fluorescence behavior is termed aggregation-induced emission (AIE) [7–10]. Organic fluorophores that possess AIE capability are called AIEgens, such as tetraphenylethylene (TPE), cyanostilbene, 9,10-distyrylanthracene (DSA), and hexaphenylsilole (HPS). Taking HPS as an example, as is shown in Scheme 1a, when the HPS is dissolved in a good solvent (THF), the solution showed no fluorescence emission. As the fraction of $H_2O$ is gradually increased, the molecules of HPS gradually form aggregates, and the fluorescence of the solution is turned on when the $H_2O$ fraction reaches 80%.

Supramolecular self-assembly [11–14] provides a great power for chemists to organize molecular building blocks into highly ordered entities through various non-covalent interactions. These include macrocyclic host–guest interactions [15–22], quadruple hydrogen bonding [23–32], π–π stacking interactions [33–39], hydrophobic interactions [40–44],

and metal/anion coordination interactions [45–48]. By combining the advantages of supramolecular self-assembly and AIE, a new strategy for generating fluorescence emerged: supramolecular-self-assembly-induced emission (SAIE). By linking supramolecular binding units onto AIE fluorophores, these molecules can aggregate in an ordered way instead of disordered aggregation by introducing a poor solvent, resulting in highly emissive nano-assemblies. For example, Yang and colleagues reported an supramolecular-assembly-induced yellow emission driven by a pillar[5]arene-based host–guest interaction [49]. In their system, they employed DSA as a bridge to link two pillar[5]arene macrocycles to produce DSA-bridged bis(pillar[5]arene)s, i.e., DSA-(P5)2. This supramolecular self-assembly unit can complex with a triazole-based neutral linker (NG2) via a pillar[5]arene-based host–guest interaction, leading to the formation of luminescent supramolecular materials. The on–off, switchable luminescent behavior of the host–guest complex based on supramolecular assembly and disassembly by an AIE strategy may promote the development of a new fluorescent probe method for labeling and tracking proteins and DNA cells. They first synthesized the host and guest molecules, which were fully characterized by $^1$H NMR, $^{13}$C NMR, and MALDI-TOF MS spectroscopy. The experimental results showed that the covalent grafting of artificial macrocyclic host molecules to DSA bridges maintained the remarkable AIE behavior of DSA cores, offering a new macrocyclic assembling block with excellent luminescence properties. In contrast to conventional AIE behavior by the introduction of unfavorable solvents, the strong yellow fluorescence of DSA-(P5)$_2$ was realized by supramolecular self-assembly through a pillar[5]arene-based host–guest inclusion interaction. The authors claimed that it was the first example of a pillar[5]arene-based supramolecular polymer with enhanced yellow fluorescence, providing promising applications in biosensing, optical switching, and bioimaging.

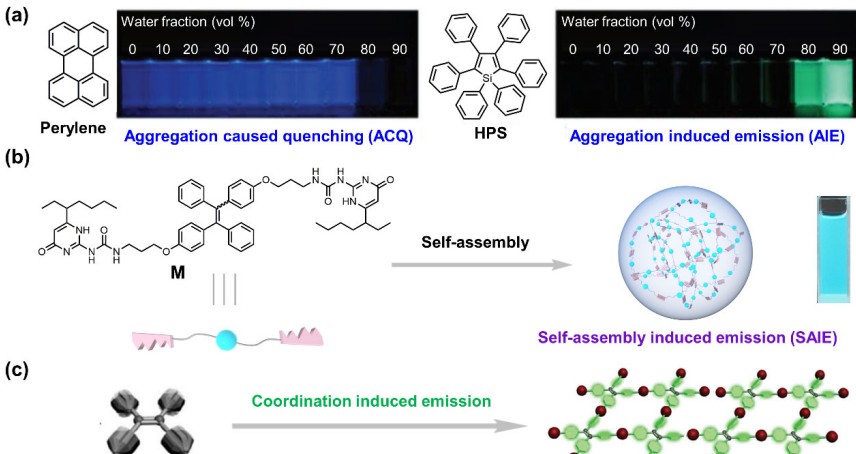

**Scheme 1.** (**a**) Chemical structures and photos of solution of perylene and HPS with different poor solvent (H$_2$O) fractions, taken under UV irradiation. (**b**) Schematic illustration of the self-assembly-induced emission (SAIE) based on the ditopic ureidopyrimidinone monomer M. (**c**) Cartoon representation of coordination-induced emission.

Our group also developed a series of emissive nanoparticles for light harvesting by the supramolecular self-assembly of AIE monomers [50–58]. As is shown in Scheme 1b, we synthesized a TPE group-bridged ditopic ureidopyrimidinone (UPy) derivative, M [58]. UPy-based quadruple hydrogen bonding bears a high binding constant (K$_{dimer}$ > 10$^7$ M$^{-1}$ in pure chloroform). The TPE group provides the compound M with excellent AIE behavior, and the UPy moieties endow it with quadruple-hydrogen-bonding capability. Additionally, supramolecular polymeric nanoparticles based on M were self-assembled through a mini-emulsion method with the help of cetyltrimethyl ammonium bromide (CTAB), resulting in a highly emissive solution. Compared with a disordered AIE aggregate formed by the introduction of poor solvents, these nanoaggregates formed by supramolecular self-assembly

are more orderly and structured, resulting in a higher fluorescence emission efficiency. The solution of M in its nanoparticle form demonstrated a bright cyan fluorescence under a UV lamp at 365 nm (Scheme 1b). By contrast, the solution of M in dichloromethane showed no fluorescence under a UV lamp at 365 nm. This phenomenon was also verified by fluorescence spectra. Moreover, the absolute fluorescence quantum yield of the M nanoparticles was determined to be 80.12%. An efficient, artificial light-harvesting system in water was then fabricated by using Nile red (NiR) as an energy acceptor in the Förster resonance energy transfer (FRET) process, which exhibited a high antenna effect at a high D/A (donor to acceptor) ratio and good energy-transfer efficiency.

Metal coordination interactions are often employed to construct various supramolecular assemblies such as macrocylces, cages, and frameworks. Similar to host–guest interactions and the quadruple hydrogen bonding mentioned above, a metal coordination interaction can also induce the strong emission of AIE building blocks. As is shown in Scheme 1c, the immobilization of functionalized TPE within rigid, porous metal–organic frameworks (MOFs) by metal coordination could turn on fluorescence emission in the typically non-emissive TPE units [59]. Generally, short, intermolecular TPE contacts are the reason for the turn-on fluorescence. In this work, the authors proved that the tight packing of the TPE fluorophores is not necessary for turn-on emission. They demonstrated that fixing AIE-type fluorophores to metal ions within a rigid matrix could be an alternative method to restrict the motion of the phenyl groups. The coordination of tetrakis(4-carboxyphenyl)ethylene ($TCPE^{4-}$) to $d^{10}$ metal cations affords fluorescent MOFs. $TCPE^{4-}$ was incorporated in a MOF matrix by reacting $H_4TCPE$ with $Zn(NO_3)_2 \cdot 6H_2O$ in a solution of ethanol and N,N-diethylformamide (DEF) at 75 °C. Yellow block crystals of $Zn_2(C_{30}H_{16}O_8)(H_2O)_2 \cdot 4DEF$ were synthesized by this reaction. An X-ray crystal analysis showed staggered, 2D sheets produced from the paddlewheel-shaped $Zn_2$-$(O_2C\text{-})_4$ and secondary building units bridged by $TCPE^{4-}$ ligands. It is worth noting that these TPE units are not in van der Waals contact, yet they show fluorescence lifetimes similar to those of molecular aggregates (Scheme 1c). Although the tightly packed TPE cores are absent, which are necessary for turn-on fluorescence in AIE materials, the MOFs are fluorescent. Additionally, the novel obtained MOFs are porous and demonstrate guest-dependent fluorescence emission owing to the spatial isolation of the TPEs, resulting in potential applications in sensing. The MOF samples showed fluorescent responses towards different solvents: a blue shift from 467 to 457 nm occurred after exposure to ethylenediamine, whereas red shifts of 6 and 10 nm were observed upon exposure to cyclohexanone and acetaldehyde, respectively.

Aside from metal coordination, anion coordination also plays a pivotal role in supramolecular chemistry. Furthermore, anion coordination chemistry has received considerable interest recently due to the extensive applications of anions in many areas, such as in the environment, biology, medicine, and functional materials [60–62]. Therefore, in this minireview, we aim to summarize the recent developments of anion-complexation-induced emission (ACIE) materials constructed from AIE fluorophores. The combination of anionic coordination chemistry and AIE science opens a new window for the manufacture of organic, luminescent materials. For example, in the field of anion sensing, traditional ACQ-based sensors often exhibit a "turn-off" mode when they are bound to anions. In contrast, AIE-based sensors generally perform a fluorescence "turn-on" mode when combined with anions, greatly improving detection sensitivity. It is worth noting that we herein focus mainly on systems based on small, inorganic anions. Considering the different anion receptors, the contents are divided as follows: (1) AIEgen decorated with a urea group as a ligand for ACIE, (2) AIEgen decorated with an imidazolium group as a ligand for ACIE, (3) AIEgen decorated with a dimethylformamidine group as a ligand for ACIE, and (4) AIEgen decorated with other groups as ligands for ACIE.

## 2. AIEgen Decorated with a Urea Group as a Ligand for ACIE

Phosphates play vital roles in many biological processes, such as in bone mineralization, DNA replication, and cellular signaling [63]. Inorganic phosphates are important for the homeostasis of phosphate in the body and are involved in many enzymatic reactions. As a result, the investigation of phosphate-anion complexation is crucially important for understanding biological activities. Recently, Wu and colleagues discovered that the oligourea ligand exhibits excellent coordination capability towards phosphate anions [64]. To promote the application of oligourea–phosphate coordination in luminescent materials, Wu and colleagues further designed and synthesized two tetrakis(bisurea) compounds, **L1** and **L2**, in which four bisurea groups are linked to a TPE core (Figure 1) [65]. Firstly, they chose the nitrophenyl-substituted compound **L1** for the investigation owing to its stronger binding ability with phosphate anions. However, the weak emission of **L1** was observed upon aggregation, possibly due to the electron-withdrawing property of these nitro groups.

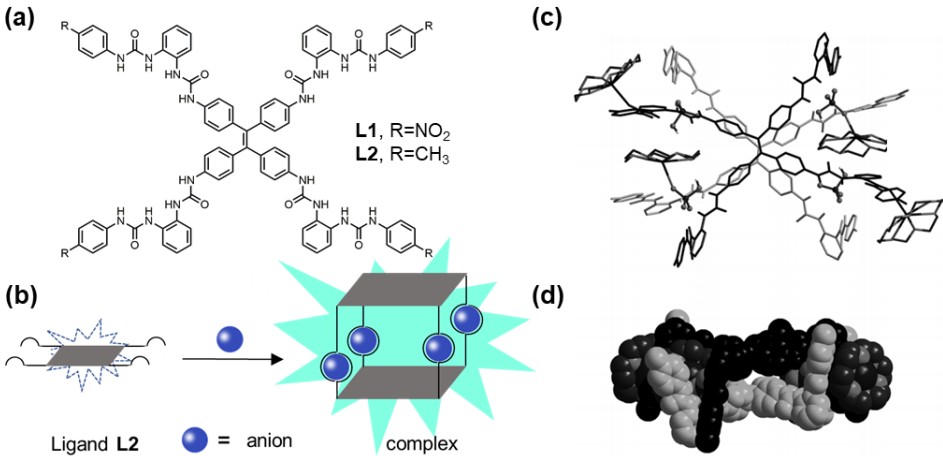

**Figure 1.** (**a**) Chemical structure of L, (**b**) schematic illustration of the anion-coordination-induced emission, (**c**) crystal structure of the complex, and (**d**) side view of the space-filling representation. Reproduced with permission from ref. [65]. Copyright 2014, Wiley Publishers.

Then they studied the methyl-group-substituted compound **L2**, which showed no fluorescence in a dilute solution but displayed a strong emission at 500 nm by adding a poor solvent, indicating that **L2** is a typical AIE molecule. Intriguingly, the non-emissive **L2** in a dilute solution ($1 \times 10^{-5}$ M) started to fluoresce in the presence of phosphate anions. When 10 equiv. of $PO_4^{3-}$ or 16 equiv. of $HPO_4^{2-}$ anions were introduced into a DMSO solution of **L2**, the emission maxima were achieved, with enhancements of approximately 27- and 20-fold, respectively (Figure 2). The authors also investigated other anions, such as $SO_4^{2-}$, $NO_3^-$, $CO_3^{2-}$, $Br^-$, $F^-$, $Cl^-$, $ClO_4^-$, $I^-$, $AcO^-$, and $HSO_4^-$. However, all these anions exhibited worse emission enhancement in contrast to phosphate anions. To achieve further insight into the coordination parameters of **L2** with $HPO_4^{2-}$, a single-crystal structure of the complex was analyzed (Figure 1c,d). The complex displayed a 4:2 (anion/ligand) ratio, in which the four $HPO_4^{2-}$ anions were clamped by two **L2** molecules stacking in a face-to-face mode. These anion coordination forces greatly rigidify the four rims of **L2** and further restrict the intramolecular motion of the TPE groups, leading to a remarkable emission of the complex.

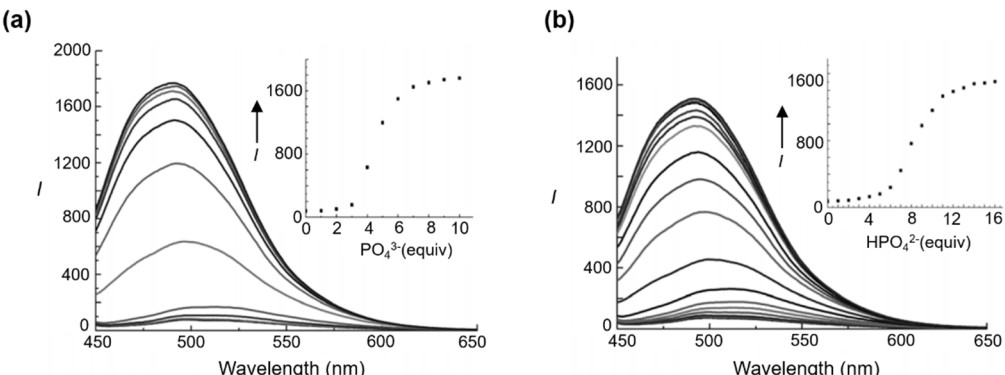

**Figure 2.** Fluorescence titration spectra of **L2** (10 mM in DMSO) upon addition of (**a**) 0–10 equiv. of $PO_4^{3-}$, and (**b**) 0–16 equiv of $HPO_4^{2-}$. Insets: The increase of fluorescence intensity at 494 nm. Reproduced with permission from ref. [65]. Copyright 2014, Wiley Publishers.

In a follow-up study, the same group reported a bis–bis(urea)-decorated TPE molecule [66]. The fluorescence of this molecule could also be enhanced in the presence of phosphate anions. Moreover, the emission of the anion complex exhibited a second enhancement accompanied by a red shift upon the addition of methyl viologen. This should be due to the electron-accepting nature of the methyl viologen group, which is prone to complex with electron-rich systems. In addition to AIEgen, Haley, Johnson, and colleagues constructed an "off–on" fluorescent sensor based on 2,6-bis(2-anilinoethynyl)pyridine bisureas for the detection of chloride in water [67].

Urea-functionalized TPE derivatives can also be used as anion probes. Pigge and colleagues synthesized a series of four alkyl or aryl urea groups attached to TPEs (Figure 3a) [68]. In the presence of monovalent anions such as halide, carboxylate, nitrate, and azide, these TPE-based derivatives exhibited enhanced fluorescence due to the aggregation of the TPEs through the urea–anion hydrogen bonding interaction. The authors investigated the complexation capabilities between these three TPE derivatives and the monovalent anions, indicating that the anion basicity plays an important role in emission enhancement. A fluoride anion was proved to be the best anion to enhance fluorescence. As is shown in Figure 3b, the enhanced emission of **L5** upon the addition of TBAF could also be detected visually under UV irradiation. This work demonstrates the feasibility of TPE-derived fluorescent anion probes/detectors.

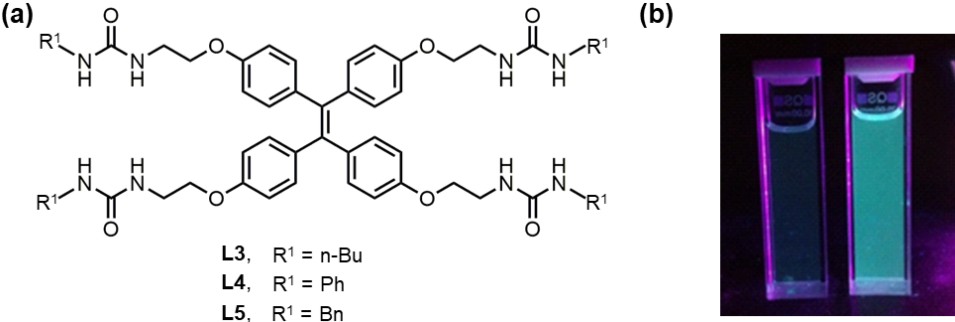

**L3**, $R^1$ = n-Bu
**L4**, $R^1$ = Ph
**L5**, $R^1$ = Bn

**Figure 3.** (**a**) Chemical structure of TPE-ureas **L3**, **L4**, and **L5** and (**b**) fluorescent images of urea **L5** (7.6 mM) in 0.15% DMSO/THF (**left**) and an identical solution containing 4.5 equiv. TBAF (**right**). Reproduced with permission from ref. [68]. Copyright 2014, Elsevier Publishers.

## 3. AIEgen Decorated with an Imidazolium Group as a Ligand for ACIE

Among the key anions in biological systems, pyrophosphate anion (PPi) has an important role in many metabolic processes. However, excess PPi in an environment may cause eutrophication problem. Therefore, the design and preparation of PPi detectors with a high sensitivity and selectivity is highly desired. It is noteworthy that organic sensors for

PPi recognition in aqueous media are not well-documented. Fluorescent, organic nanoparticles constructed via supramolecular self-assembly have received much interest owing to their low cost and cytotoxicity and good biocompatibility and photostability. In this context, Cao, Kim, and colleagues reported a new bis-imidazolium-derived TPE probe (BIM-TPE) for sensing PPi (Figure 4) [69]. The bis-imidazolium group was used as the anion receptor for their specific $(CH)^+ \cdots$ phosphate anion interaction, while the TPE units were used as the fluorescent signal group. They demonstrated that the BIM-TPE exhibited a "turn-on" emission towards PPi over other anions in aqueous media. As is shown in Figure 4, the BIM-TPE formed small, sphere-shaped nano-assemblies in water with a very weak fluorescence. However, with the addition of PPi anions, they changed into large, rod-like nano-assemblies exhibiting a strong fluorescence. A combination of techniques, including $^1$H NMR, fluorescence measurements, transmission electron microscopy (TEM), and dynamic light scattering (DLS), were employed to study the binding mode between the receptor and the PPi.

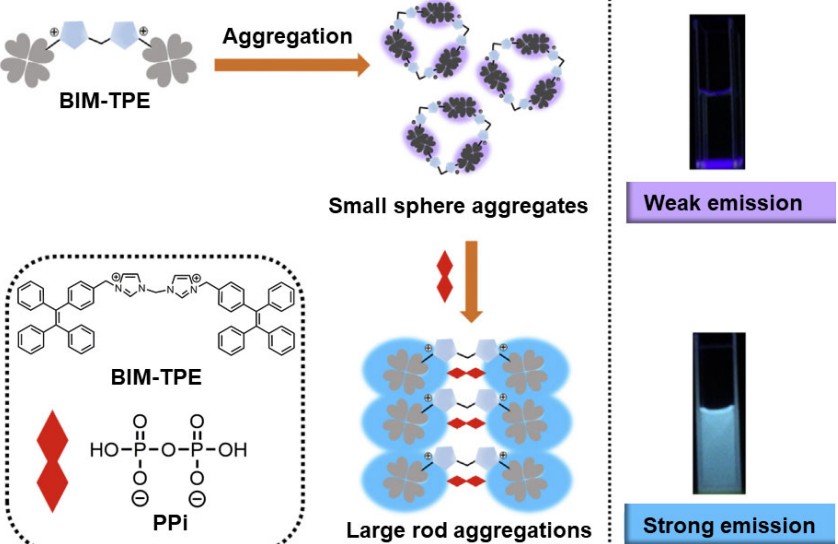

**Figure 4.** Chemical structure of BIM-TPE and schematic illustration of the detection strategy for PPi. Reproduced with permission from ref. [69]. Copyright 2017, Elsevier Publishers.

As mentioned above, phosphate anions can be complexed with ligands to trigger the fluorescence of the material. In those cases, however, the ligands are not a good sensor for the phosphate anions. Due to the interference of solvents and competition from other anions, there still remains a significant challenge in designing phosphate fluorescence sensors with high selectivity and sensitivity. A good method is to design chemical sensors with multiple complexation sites that are preorganized for specific anions. An organic macrocycle is a good candidate for such purposes. To this end, Xiong et al. employed an organic macrocycle **L6** with two bridged TPE groups and two imidazolium groups to detect PPi anions in an aqueous solution (Figure 5). In the presence of $Zn^{2+}$, **L6** exhibited high selectivity towards PPi anions and displayed ratiometric fluorescence changes.

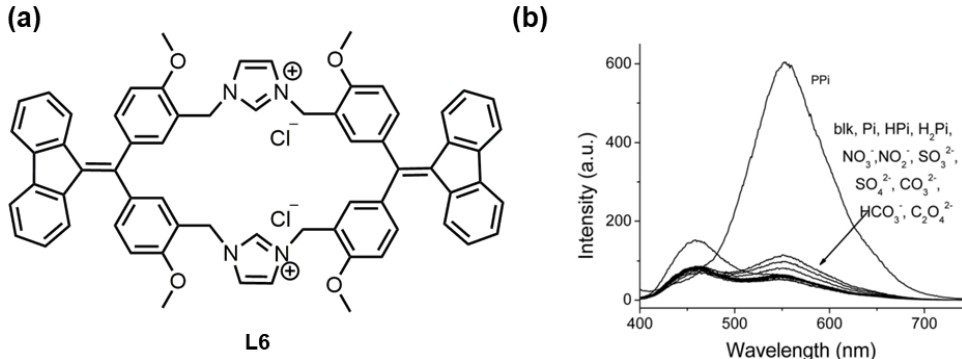

**Figure 5.** (**a**) Chemical structure of macrocyclophane **L6** and (**b**) The fluorescence spectra of **L6** in water containing 50% DMSO with addition of different anions in the presence of $Zn^{2+}$. $\lambda_{ex}$ = 378 nm. Reproduced with permission from ref. [70]. Copyright 2022, the Royal Society of Chemistry.

As one of the most important anions in the biological system, $SO_4^{2-}$ is employed in the synthesis of many related biomolecules such as lipids, glycoproteins, and mucopolysaccharides. Recently, Cao and colleagues designed and synthesized two cleft-type, TPE-based imidazoliums with 1,2−phenyl and 1,2−phenylmethyl spacers. The imidazolium groups were chosen owing to their anion binding capability [71]. The sensor with a rigid frame demonstrated a high binding ability towards $SO_4^{2-}$ as well as polyphosphate (such as PPi, ADP, and ATP, etc.) accompanied with increasing emission in aqueous media.

## 4. AIEgen Decorated with a Dimethylformamidine Group as a Ligand for ACIE

As mentioned above, anions play an important role in human health and environmental protection. Many efforts have been made to exploit new approaches to detect or sense anions. Due to the competitive, disruptive binding of water towards anions, it remains a significant challenge to develop anion sensors with a high selectivity and fluorescence turn-on mode in an aqueous solution. In 2019, Zheng and colleagues reported a TPE-based dimethylformamidine derivative as a chemical sensor, which could selectively detect $PO_4^{3-}$ in aqueous media among other anions (Figure 6) [72]. The authors first synthesized **L7** in three steps from dimethoxy TPE. The target molecule was fully characterized by $^1$H NMR, $^{13}$C NMR, IR, and HRMS spectra. Due to the iminium groups, **L7** could be dissolved in polar solvents, even water. A series of anions were used for enhancing the fluorescence of **L7**. These included sodium phosphate (Pi), sodium hydrogen phosphate (HPi), sodium dihydrogen phosphate (H2P), PPi, $Na_2SO_4$, $Na_2SO_3$, $NaNO_3$, $NaNO_2$, $Na_2C_2O_4$, NaF, NaCl, NaBr, and NaI. As a result, only $PO_4^{3-}$ (Pi) could induce a dramatic turn-on fluorescence of **L7** and the other anions showed no effect on the emission. This should be due to the interaction between **L7** and Pi, in which the electrostatic attractions might induce a decrease in solubility and the aggregation of **L7**, leading to an AIE effect. Therefore, **L7** demonstrated great potential for selectively probing $PO_4^{3-}$ anions in aqueous media through a fluorescence turn-on mode.

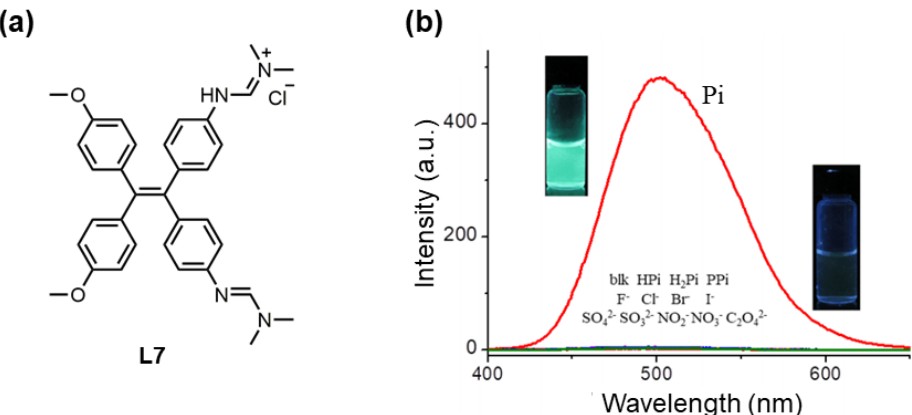

**Figure 6.** (**a**) Chemical structure of **L7** and (**b**) the fluorescence spectra of **L7** before and after addition of Pi, HPi, $H_2$Pi, Ppi, $SO_3^{2-}$, $SO_4^{2-}$, $NO_2^-$, $NO_3^-$, $C_2O_4^{2-}$, $F^-$, $Cl^-$, $B^-$, and $I^-$ in water. Inset, the photos of the solution of **L7** with and without phosphate anions, under a 365 nm light from a portable UV lamp. Reproduced with permission from ref. [72]. Copyright 2019, Wiley Publishers.

As mentioned above, the detection of $PO_4^{3-}$ is very important. However, many fluorescent sensors towards $PO_4^{3-}$ could not be soluble in pure water, which greatly impedes their application in biological areas. In 2021, He and colleagues designed and synthesized a new, turn-on fluorescent sensor, **L8**, with a TPE core and four dimethylformamidine groups (Figure 7a) [73]. The sensor could be dissolved in pure water and showed dramatic emission enhancement towards $PO_4^{3-}$. Moreover, **L8** showed a lower detection limit (LOD) of $6.43 \times 10^{-8}$ mol/L for $PO_4^{3-}$. Notably, **L8** could also be used to sense phosphate in real water samples. Finally, the emission color change of solutions further showed the semi-quantitative visual detection capability of the sensor **L8** for $PO_4^{3-}$ (Figure 7b).

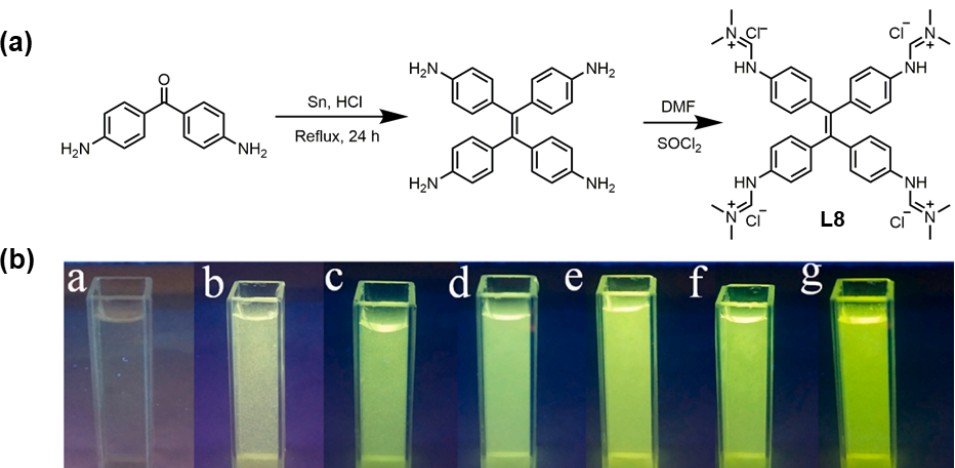

**Figure 7.** (**a**) Synthesis of sensor **L8** and (**b**) photos of the solutions of sensor **L8** ($1 \times 10^{-5}$ mol/L) without and with different concentrations of phosphate anion (from b to g, the concentrations were 50, 100, 150, 200, 250, and 300 μmol/L, respectively. $\lambda_{ex}$ = 365 nm). Reproduced with permission from ref. [73]. Copyright 2021, Elsevier Publishers.

## 5. AIEgen Decorated with Other Groups as Ligands for ACIE

With the development of supramolecular anion host–guest chemistry, halogen bonding (HB) has become a valuable supramolecular interaction in this field. Compared to hydrogen bonding, the HB interactions can usually promote anion affinity and selectivity within anion sensors. Recently, Beer, Langton, and colleagues synthesized a multidentate anion receptor, 4·XB, which contained a TPE core and several perfluoroaryl iodotriazole halogen- bonding groups (Figure 8) [74]. $^1$H NMR titration showed that the halogen-bonding geometry had a

large impact on the coordination mode and binding affinity. A combination of experiments, including fluorescence measurements, DLS, TEM, and X-ray crystal analysis, revealed the formation of aggregates induced by halogen bonding, leading to a remarkable turn-on fluorescence for chloride sensing. Similar to other TPE derivatives, fluorescence spectra of 4·XB showed that it is a weakly emissive molecule. An obvious increase in fluorescence intensity upon the addition of water ($f_w$) was observed when $f_w > 60\%$, indicating that 4·XB is a typical AIE derivative due to the TPE core. Fluorescence experiments showed that 4·XB displayed a significant increase in fluorescence intensity in the presence of chloride. By contrast, its hydrogen bonding analogue, 4·HB, showed no emission enhancement. Notably, the addition of hexafluorophosphate did not change the fluorescence spectrum of 4·XB, suggesting that the non-coordinating anion does not work and that the XB-mediated coordination to chloride is necessary. DLS data of the THF solution ($10^{-5}$ M) of 4·XB indicated that the free molecule is non-aggregated without Cl$^-$. However, the formation of nano-aggregates with a diameter of about 100 nm was observed upon the addition of 1 equiv. of TBACl.

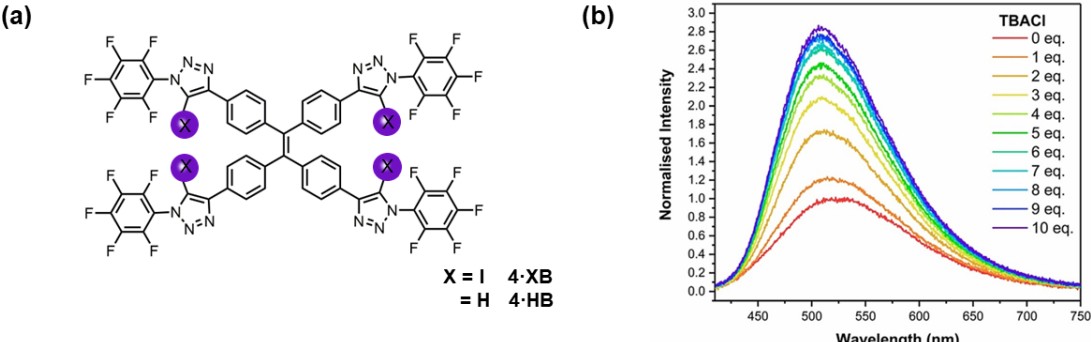

**(a)** X = I  4·XB
= H  4·HB

**Figure 8.** (**a**) Chemical structure of 4·XB and 4·HB and (**b**) fluorescence spectra (THF, $10^{-5}$ M, $\lambda_{ex} = 350$ nm) of 4·XB in the presence of increasing equivalents of TBACl. Reproduced with permission from ref. [74]. Copyright 2021, Wiley Publishers.

In another work, Zhao, Tang, and colleagues synthesized a series of pyridinium-functionalized TPE salts containing different alkyl chains. They investigated the impact of chain length on their photophysical properties [75]. Among these salts, TPEPy-1 exhibited excellent anion-complexation-induced emission behavior (Figure 9a). In detail, TPEPy-1 showed a dual-mode fluorescence "turn-on" behavior, including NO$_3^-$ and ClO$_4^-$, in an aqueous solution. As is shown in Figure 9b, when NO$_3^-$ was gradually titrated into the solution of TPEPy-1, a new fluorescence peak at 540 nm appeared and increased. Meanwhile, the colorless solution changed to bright yellow (Figure 9c). The introduction of ClO$_4^-$ into the solution resulted in a new fluorescence peak at 570 nm (Figure 9d). When 1.2 mM ClO$_4^-$ was added, the solution showed a bright orange color under 365 nm UV irradiation. It should be noted that other anions, such as CO$_3^{2-}$, H$_2$PO$_4^-$, HCO$_3^-$, HPO$_4^{2-}$, PO$_4^{3-}$, SO$_4^{2-}$, and F$^-$, Cl$^-$, Br$^-$, and I$^-$, caused negligible emission changes. SEM images showed that TPEPy-1 formed amorphous particles. By contrast, regular microcubics were formed upon the addition of NO$_3^-$, while microplates were observed in the presence of ClO$_4^-$. The weak interactions between NO$_3^-$/ClO$_4^-$ and the cationic pyridinium group and the van der Waals force through the alkyl chains should be the reason for improving the ordered organization of TPEPy-1, which dramatically enhanced the emission. Moreover, the authors designed and prepared a number of pyridinium-functionalized tetraphenylethene salt analogues with different alkyl chains. They studied the influence of the alkyl chain length on the performance of the cationic fluorophores. In contrast to TPEPy-1, analogues with longer alkyl chains resulted in a considerable enhancement in fluorescence due to the stronger hydrophobicity. The morphology of the self-assembled fluorophores can be tuned from microplates to microrods by changing the alkyl chain length, while TPEPy-1 generated a blue-shift fluorescence in contrast to other fluorophores. Additionally, TPEPy-1

displayed a unique, dual-mode turn-on fluorescence in response to $NO_3^-$ and $ClO_4^-$. This was because the anions induced the self-assembly of the fluorophores. This side chain engineering offers a good way to tune the fluorescence properties and to study the structure-property relationship.

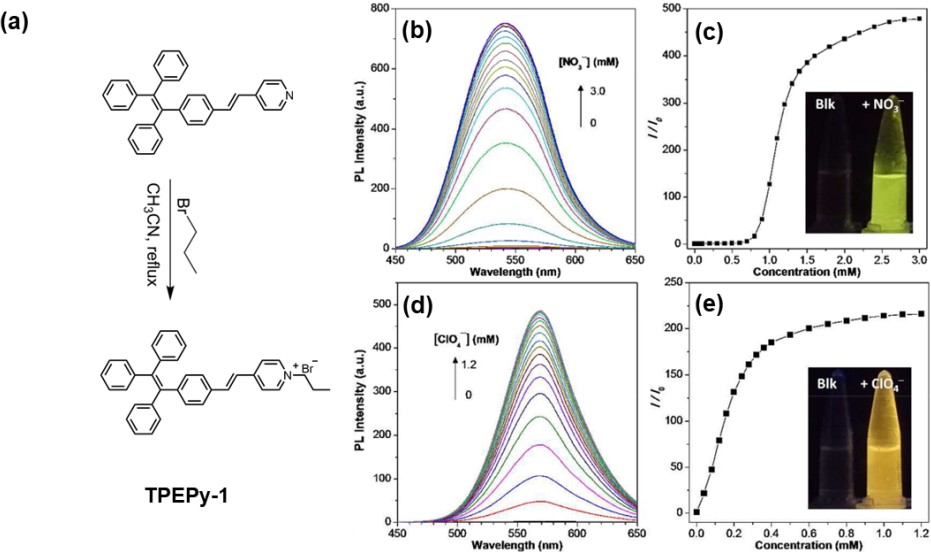

**Figure 9.** (**a**) Synthetic route of TPEPy-1. Fluorescence spectra of TPEPy-1 (20 μM) in the presence of (**b**) $NO_3^-$ and (**d**) $ClO_4^-$. Plot of relative emission intensity ($I/I_0$) at (**c**) 540 nm versus the concentration of $NO_3^-$ and (**e**) 570 nm versus the concentration of $ClO_4^-$. Inset: photographs of TPEPy-1 without and with (**c**) $NO_3^-$ or (**e**) $ClO_4^-$ under 365 nm UV irradiation. Excitation wavelength: 375 nm. Reproduced with permission from ref. [75]. Copyright 2018, American Chemical Society.

## 6. Conclusions and Perspectives

In summary, recent advances in the field of supramolecular luminescent materials based on an anion-complexation-induced emission mediated by various anion receptors are reviewed. These anion receptors cover a series of different binding sites, including the urea group, imidazolium group, dimethylformamidine group, and other anion receptors. The different molecular structures of these anion receptors endow them with the capability to complex with one or two kinds of anions, leading to a strong fluorescence enhancement of the receptors. Notably, some of these receptors can be employed as a "turn-on" fluorescent sensor owing to their high selectivity and sensitivity towards specific anions. The examples listed in this review represent the research area of functional materials/sensors mediated by anion–receptor interactions. Although there are far fewer examples of anion-coordination-induced emission than that of metal-coordination-induced emission, this area of research is crucial due to the importance of anion coordination in many aspects. Obviously, the description in this paper should not be a limitation but an inspiration for future study. We believe an increasing number of anion-complexation-induced emission materials based on different anions and receptors will be developed in the near future, owing to their fascinating features.

**Author Contributions:** Conceptualization, T.X.; Writing—original Draft Preparation, D.R. and T.X.; Writing—review and Editing, D.R., L.Z., H.Q. and T.X.; Supervision, T.X. All authors have read and agreed to the published version of the manuscript.

**Funding:** T.X. acknowledges the National Natural Science Foundation of China (No. 21702020). L.Z. and H.Q. acknowledge the Postgraduate Research & Practice Innovation Program of Jiangsu Province (grant No. KYCX22_3012 and KYCX22_3015).

**Data Availability Statement:** Not applicable.

**Conflicts of Interest:** The authors declare no conflict of interest.

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
