# Peer review of "Anion-Complexation-Induced Emission Based on Aggregation-Induced Emission Fluorophore"

_chemistry, doi:10.3390/chemistry5010019_

Round 1

Reviewer 1 Report

The manuscript by Xiao and coworkers describes sensors based on aggregation-induced emission for the detection of anions. The topic of the review is relevant but the organization of the review and the examples included need to be improved in a major revision. Suggested changes include the following:

The title should explain the meaning of "AIEgen"

The first 3 paragraphs have 57 references and the rest of the review has only 13 references. This is a major problem as only 16 examples are fully described in the review. This should be improved and more examples describing the corresponding chemical structures and sensing properties need to be included. For this, some of the 54 references in the introduction should be explained in detail. Additional references should be also included to make the review more comprehensive, and therefore, more relevant to the field.

Another important aspect is that 21 references are self-citations of the corresponding author (searching "Xiao, T" in the reference list). Considering that the review has a total of 70 references, this is 30% of self-citations. This should be revised to only include the relevant self-citations to the review (note that all self-citations are in the first 3 paragraphs, and therefore, no examples are fully described)

Additionally, the introduction (the first 3 paragraphs) should have several figures describing the chemical structures of the examples described.

The introduction should also put in context the advantages of aggregation-induced emission sensors for anions versus other sensors for anions, to give to the reader the advantages and drawbacks of these systems.

Reviewer 2 Report

In the review, the authors summarize the anion coordination induced AIE structures systematically. This work enlightens the development trend of anion complexes based AIE and has important reference significance. This review article is comprehensive, up to-date, and easy to follow. I recommend it for publication in Chemistry after minor revision according to the suggestions as follow:

1. Please revise the following sentences.

Line 7: Aggregation-induced emission (AIE) materials have attracted increasing research interest 7 in recent years owing …

Line 48: … opened a new window to …

Line 145: As ….

2. The conclusion and perspectives part is dry. Author could add some comparisons of different anion groups, such as providing the pros and cons for each stimuli.

3. Some references should be added.

10.1039/B105159H 

10.1021/acs.chemrev.5b00263

10.3390/molecules24152711

10.3390/molecules27227881

Reviewer 3 Report

This work proposes a short review on the anion-complexation-induced emission based on AIEgen. This topic is very interesting, since normally anion binding quenched the fluorescence of the host via PET, the supramolecular assembly based on the AIE-based materials is thus important. the author has not contributed to this research area (ref. 46-54), but they have included recent and representative references.  Even so, the reviewer thinks the overall organization of this review is poor, it might be publishable after addressing the concerns and comments outlined below:

1. The author should discuss more about the merits of anion coordination-driven assemblies of AIEgens, like the turn-on” fluorescent sensors for anions. This sentence is only shown in the conclusion (line 257).

2.    For the title in each category, ‘emission induced by anion-…’ are unclear. Do the authors want to convey that the urea/ imidazolium/ dimethylformamidine/other motifs functionalized AIE-based fluorophores/sensors/ligands?

3. In each section, the author repeatedly stressed the importance of some anions, for instance, the phosphates (lines 58-61; lines 162-163), and sulfates (lines 145-147; lines 162-163) are mentioned more than once. 

Round 2

Reviewer 1 Report

The authors have replied to all my suggested corrections, improving the quality of the manuscript. Therefore, I recommend the publication of the review article after a very minor correction:

The only change that I would suggest to the present version is in the title: To change "AIE" to "Aggregation-Induced Emission" for clarity. This will help to search the paper as the acronym AIE is difficult to understand for a non-specialized public

Reviewer 3 Report

The authors have largely revised the manuscript.

Additional comments:

1. The author somewhat overstated their work in the introduction part, please make it concise.

2. Agree with what author said. I still think the subtitles for each section can be more specific, it is still unclear to me. Apparently, these AIEgen were functionalized by specific functional groups.
